# Tackling public health data gaps through Bayesian high-resolution population estimation: A case study of Kasaï-Oriental, Democratic Republic of the Congo

Gianluca Boo[1]*, Edith Darin[2], Heather R. Chamberlain[1], Roland Hosner[3], Pierre K. Akilimali[4], Henri Marie Kazadi[5], Chibuzor C. Nnanatu[1], Attila N. Lázár[1], Andrew J. Tatem[1]

**1** WorldPop, School of Geography and Environmental Science, University of Southampton, Southampton, United Kingdom, **2** Nuffield Department of Population Health, University of Oxford, Oxford, United Kingdom, **3** Flowminder Foundation, Stockholm, Sweden, **4** École de Santé Publique de Kinshasa, Université de Kinshasa, Kinshasa, Democratic Republic of the Congo, **5** Institut National de la Statistique, Bureau Central du Recensement, Kinshasa, Democratic Republic of the Congo

\* gianluca.boo@soton.ac.uk

## Abstract

Most low- and middle-income countries face significant public health challenges, exacerbated by the lack of reliable demographic data supporting effective planning and intervention. In such data-scarce settings, statistical models combining geo-located survey data with geospatial datasets enable the estimation of population counts at high spatial resolution in the absence of dependable demographic data sources. This study introduces a Bayesian model jointly estimating building and population counts, combining geolocated survey data and gridded geospatial datasets. The model provides population estimates for the Kasaï-Oriental province, Democratic Republic of the Congo (DRC), at a spatial resolution of approximately one hectare. Posterior estimates are aggregated across health zones and health areas to offer probabilistic insights into their respective populations. The model exhibits a –0.28 bias, 0.47 inaccuracy, and 0.95 imprecision using scaled residuals, with robust 95% credible intervals. The estimated population of Kasaï-Oriental for 2024 is approximately 4.1 million, with a credible range of 3.4 to 4.8 million. Aggregations by health zones and health areas reveal significant variations in population estimates and uncertainty levels, particularly between the provincial capital, Mbuji-Mayi and the rural hinterland. High-resolution Bayesian population estimates allow flexible aggregation across spatial units while providing probabilistic insights into model uncertainty. These estimates offer a unique resource for the public health community working in Kasaï-Oriental, for instance, in support of a better-informed allocation of vaccines to different operational boundaries based on the upper bound of the 95% credible intervals.

**Data availability statement:** The input data and scripts used to develop the high-resolution population estimates, as well as the grid-cell posterior distributions and scripts to aggregate them within user-defined geographic boundaries are available at https://doi.org/10.5281/zenodo.15113660.

**Funding:** The micro-census data collection was performed during the "GRID3 Mapping for Health" project funded by Gavi, the Vaccine Alliance (RM-86720420A2) awarded to AJT. The high-resolution population estimates were produced as part of the "GRID3 – Phase 2 Scaling" project funded by the Bill & Melinda Gates Foundation (INV-044979) awarded to AJT. Funders did not have any role in the study design, data collection and analysis, decision to publish, and preparation of the manuscript.

**Competing interests:** The authors have declared that no competing interests exist.

## Introduction

The effectiveness of public health programmes in low- and middle-income countries is impacted by various factors, including limited financial and human resources, logistical constraints, insecurity, and restricted access to healthcare facilities [1]. Among these issues, the lack of actionable national and sub-national data on population sizes and demographic characteristics is an additional barrier to effective planning and intervention [2,3]. In particular, the absence of recent, granular demographic denominators hampers efforts, such as disease surveillance, vaccination campaigns, and resource allocation in countries that arguably need them the most [4]. For instance, the inability to effectively allocate mosquito protection measures, such as bed nets, in remote rural areas reduces malaria transmission control [5]. Similarly, in densely populated urban slums, the lack of reliable demographic baselines undermines efforts to monitor tuberculosis outbreaks and provide adequate diagnostic and treatment capabilities [6]. Furthermore, during the COVID-19 pandemic, the lack of robust population data severely hampered the deployment of pharmaceutical and non-pharmaceutical interventions, contributing to substantial excess mortality [7].

Traditional national and sub-national population and demographic data include population censuses, vital statistics systems, administrative records, household surveys, community-based monitoring, and local knowledge [8]. In countries where these data sources are outdated, unreliable, or non-existent, new approaches have emerged that combine geolocated household surveys with geospatial datasets derived from satellite imagery and open sources to produce high-resolution population estimates [9]. These approaches typically employ Bayesian statistical models to produce population estimates together with uncertainty measures within grid cells of approximately one hectare, which can be aggregated into larger geographic regions [10]. Recent advances have improved model accuracy by integrating settlement and building footprint data extracted from satellite imagery [11]. However, despite growing concerns about biases in the automatic detection of settlement and building features [12], existing Bayesian population models have not explicitly accounted for uncertainty in these crucial data sources [10,11,13].

In this context, the Democratic Republic of the Congo (DRC) faces significant public health challenges, exacerbated by the difficulty of reaching its most vulnerable populations because of the unavailability of reliable national and sub-national demographic data [14–16]. In particular, the last national census was completed in 1984, and subsequent demographic projections have yielded inconsistent country-level population estimates, ranging from 73.7 to 103.5 million [14]. To address this knowledge gap, we developed a Bayesian model for high-resolution population estimation in the Kasaï-Oriental province, DRC. The model leverages complete building and population counts within small and well-defined enumeration areas, referred to as micro-census clusters, to produce up-to-date gridded population estimates across the province. Grid-cell posterior distributions are aggregated and summarised within health zones and health areas to provide probabilistic insights into their respective populations. These data, developed in support of a vaccination campaign

coordinated by GRID3 [17], offer actionable demographic denominators in support of the public health community working in the province.

## Materials and methods

### Input data

We produced gridded population estimates for Kasaï-Oriental based on administrative boundaries publicly available on OpenStreetMap [18]. At the time of this study, the DRC government had not publicly released official boundaries for the 26 provinces established in 2015, and existing unofficial versions exhibited notable spatial inconsistencies. To address these discrepancies, we applied a four-kilometre buffer to the OpenStreetMap boundaries to define our modelling environment. As a result, the datasets presented below may also include small portions of the neighbouring provinces.

**Micro-census survey data.** We obtained building and population counts for 213 micro-census clusters surveyed between 1 March and 30 April 2021 in the GRID3 Mapping for Health project [19,20]. The clusters were selected using stratified sampling based on settlement type [21]. The sampling frame was created using the gridEZ algorithm [22], which defined clusters comprising approximately 80 building footprints with a maximum extent of six settled hectares. Although the number of buildings per micro-census cluster was generally consistent, the spatial extent varied between 1 and 769 grid cells due to settlement patterns and the aggregation process of the gridEZ algorithm. Additional details on the micro-census data are available in the supporting information for previous population modelling work [21].

Building counts include permanent freestanding structures within the cluster boundaries, regardless of their use and status. Population counts refer to the "de jure" population, where the person either spent at least six of the previous twelve months in the household or was expected to stay for the following six months and had spent the previous night in the surveyed household. For households where the respondent refused to be interviewed, imputations were performed using the average number of residents per household within the micro-census cluster. Six micro-census clusters in the Lomami province were added to the 207 in Kasaï-Oriental because they fell within the modelling environment. Fig 1 shows the spatial distribution of the observed building and population counts at the micro-census cluster level, categorised by settlement class.

**Gridded geospatial datasets.** We retrieved 292 open-access gridded geospatial datasets at a spatial resolution of 3 arc-seconds, approximately 100m, covering the extent of the modelling environment. Of these, 289 datasets—including land use, night-time light intensity, distance to main roads, and distance to healthcare facilities—are traditionally used to model population distributions [23]. Some datasets represent the same feature from different providers (e.g., Microsoft, Google, and OpenStreetMap), with different spatial properties (e.g., length, area, and number of nodes), summary statistics (e.g., average, sum, and empirical standard deviation), and temporal coverage. A complete list of the gridded datasets is provided in S1 Table. We also accessed three datasets describing up-to-date settlement characteristics, namely building count and area, as well as settlement class. Settlement class data comprised three broad settlement classes—urban, peri-urban, and rural—extracted from the Global Human Settlement Model Grid (GHS-MOD) [24]. Building count and area data were derived from settlement data produced by the Center for International Earth Science Information Network (CIESIN) [25], which combined various data sources (e.g., Microsoft and Google building footprints). We used the extent of the settlement data to subset the gridded datasets presented earlier to constrain our model to settled grid cells only.

We summarised the gridded datasets within the micro-census cluster boundaries using: i) the mode for the settlement class data, ii) the weighted sum based on the fraction of the pixel intersecting the boundaries for building count and area from settlement data, and iii) the average for the remaining datasets. These summaries were first compared with the original settled subsets using descriptive statistics (i.e., minimum, maximum, average, and empirical standard deviation) to detect missing or spurious values. Relationships between the summaries and both log-building counts and log-population densities (i.e., people per building) were subsequently assessed using scatterplots and Pearson correlation tests to

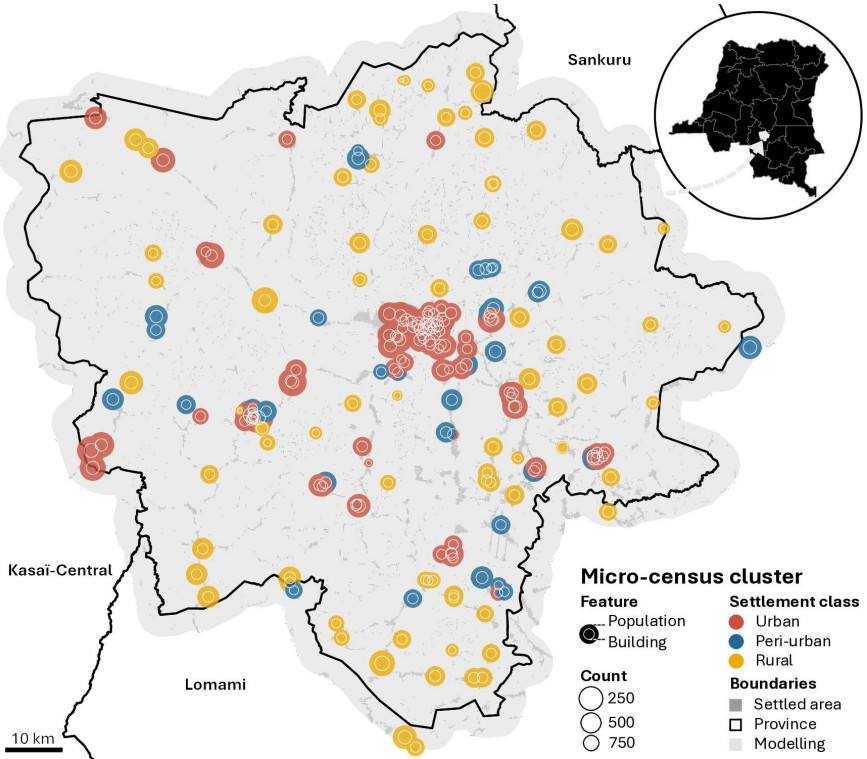

**Fig 1. Building and population counts observed in the 213 micro-census clusters located within the modelling environment categorised by settlement class.** Administrative boundaries sourced from OpenStreetMap (https://planet.osm.org) under the Open Database License (ODbL) (https://www.openstreetmap.org/copyright).

evaluate their relevance as model covariates. The summaries were checked for cross-correlations using Pearson correlation tests to detect potential multicollinearity. This exploratory analysis enabled us to select five gridded datasets exhibiting significant correlations ($p < 0.05$) with the outcome variables while remaining significantly ($p < 0.05$) uncorrelated with each other. These datasets, shown in Table 1, were included as covariates in the two model components described below.

The covariates were centred and scaled based on the average and empirical standard deviation computed at the grid-cell level. The settlement class informed some of the model parameters presented below. Data processing and covariate selection were conducted in R (4.4.0) [28], using the tidyverse (2.0.0) [29], sf (1.0-17) [30], and terra (1.7-78) [31] packages.

## Statistical model

We extended existing Bayesian hierarchical models for high-resolution population estimation [10,11,13] by incorporating observed building and population counts from micro-census survey data. This approach leverages the flexibility of the hierarchical modelling framework to incorporate observational data and propagate model uncertainty through Bayesian credible intervals (CIs) from the micro-census model to the gridded predictions and subsequent areal aggregations within larger geographic regions.

**Micro-census model.**

**Population count**  We assumed that the observed population count $P_i$ in the $i = 1, \ldots, 213$ micro-census clusters follows a Poisson distribution with rate derived from the product of the modelled building count $\lambda_i^{(B)}$ and the modelled

**Table 1. The gridded geospatial datasets implemented as covariates in the building count and population density model components.**

| | Reference year | Model component | Label |
|---|---|---|---|
| Building count from settlement data [25] | 2024 | Building count | A |
| Distance to main roads [18] | 2023 | Population density | B |
| Average dry matter productivity [26] | 2022 | Population density | C |
| Distance to violence against civilians [27] | 2022 | Population density | D |
| Terrain slope [26] | 2020 | Population density | E |

population density (i.e., people per building) $\lambda_i^{(D)}$, as shown in Equation (1). This decomposition allowed us to explicitly account for variations in population associated with the modelled building count and modelled population density rather than observed data, while incorporating different covariates and distributional assumptions for each component.

$$P_i \mid \lambda_i^{(B)}, \lambda_i^{(D)} \sim \text{Poisson}\left(\lambda_i^{(B)} \cdot \lambda_i^{(D)}\right)$$

(1)

**Building count**  Similar to Equation (1), in Equation (2), we modelled the observed building count $B_i$ in the same micro-census clusters using a Poisson distribution $\lambda_i^{(B)}$.

$$B_i \mid \lambda_i^{(B)} \sim \text{Poisson}\left(\lambda_i^{(B)}\right)$$

(2)

To account for overdispersion in the Poisson process, in Equation (3), we modelled $\lambda_i^{(B)}$ using a log-normal distribution. $\mu_i^{(B)}$ represents the expected building count on the logarithmic scale, and $\sigma_s^{(B)}$ is a settlement-class-specific variance term capturing residual variation.

$$\lambda_i^{(B)} \mid \mu_i^{(B)}, \sigma_s^{(B)} \sim \text{logNormal}\left(\mu_i^{(B)}, \sigma_s^{(B)}\right)$$

(3)

In Equation (4), we defined $\mu_i^{(B)}$ using a linear regression. $\alpha_s^{(B)}$ represents settlement-class-specific intercepts, $X_i^{(B)}$ is the covariate log-transformed building count derived from settlement data (Covariate A), $\beta_s^{(B)}$ denotes settlement-class-specific effects for the covariate. In this setup, $\alpha_s^{(B)}$ adjusts for settlement-class-specific additive bias, while $\beta_s^{(B)}$ adjusts for settlement-class-specific multiplicative bias $X_i^{(B)}$.

$$\mu_i^{(B)} = \alpha_s^{(B)} + \beta_s^{(B)} \cdot X_i^{(B)}$$

(4)

We specified weakly informative priors for $\alpha_s^{(B)}$ and $\sigma_s^{(B)}$, following a normal distribution for the former and a zero-truncated normal distribution for the latter, and a non-informative prior for $\beta_s^{(B)}$ as shown in Equation (5). These priors encode minimal prior information about the parameters, allowing the data to dominate posterior inference via the likelihood.

$$\alpha_s^{(B)} \sim \text{Normal}(0,10)$$

(5)

$$\sigma_s^{(B)} \sim \text{Normal}^+(0,10)$$

$$\beta_s^{(B)} \sim \text{Normal}(0,\infty)$$

**Population density** Similar to Equation (3), in Equation (6), we modelled population density $\lambda_i^{(D)}$ in the micro-census clusters using a log-normal distribution. $\mu_i^{(D)}$ represents the expected population density on the logarithmic scale, and $\sigma_s^{(D)}$ is a settlement-class-specific variance term capturing residual variation.

$$\lambda_i^{(D)} \mid \mu_i^{(D)}, \sigma_s^{(D)} \sim \text{logNormal}\left(\mu_i^{(D)}, \sigma_s^{(D)}\right)$$

(6)

In Equation (7), we modelled $\mu_i^{(D)}$ using a linear regression, similar to Equation (4). $\alpha_s^{(D)}$ denotes settlement-class-specific intercepts, $X_i^{(D)}$ represents the covariates distance to main roads, average dry matter productivity, distance to violence against civilians, and terrain slope (Covariates B-E), and $\beta^{(D)}$ indicates fixed effects for the covariates. Although settlement-class-specific effects were initially considered, they were ultimately discarded due to convergence issues.

$$\mu_i^{(D)} = \alpha_s^{(D)} + X_i^{(D)} \cdot \beta^{(D)}$$

(7)

In Equation (8), we specified weakly informative priors following a normal distribution for $\alpha_s^{(D)}$ and a zero-truncated normal distribution for $\sigma_s^{(D)}$, as well as a non-informative prior for $\beta^{(D)}$, reflecting the considerations in Equation (5).

$$\alpha_s^{(D)} \sim \text{Normal}(0,10)$$

(8)

$$\sigma_s^{(D)} \sim \text{Normal}^+(0,10)$$

$$\beta^{(D)} \sim \text{Normal}(0,\infty)$$

**Gridded predictions.** We leveraged the parameters $\alpha$, $\sigma$, and $\beta$ estimated at the micro-census-cluster level along with the gridded datasets to predict the population count $P_j$ for the $j = 1, \ldots, 81,446$ settled grid cells across Kasaï-Oriental. In Equation (9), $P_j$ is drawn from a Poisson distribution, with the rate parameter defined by the product of predicted building count $B_j$ and predicted population density $D_j$, as shown in Equation (1).

$$P_j \mid \lambda_j^{(B)}, \lambda_j^{(D)} \sim \text{Poisson}\left(\lambda_j^{(B)} \cdot \lambda_j^{(D)}\right)$$

(9)

Similar to Equation (3), in Equation (10), we drew $\lambda_j^{(B)}$ from a log-normal distribution. Here, $\mu_j^{(B)}$ represents the predicted expected building count per grid cell, and $\sigma_{s_j}^{(B)}$ is the settlement-class-specific variance term estimated at the micro-census-cluster level for each settled grid cell $j$.

$$\lambda_j^{(B)} \mid \mu_j^{(B)}, \sigma_{s_j}^{(B)} \sim \text{logNormal}\left(\mu_j^{(B)}, \sigma_{s_j}^{(B)}\right)$$

(10)

In Equation (11), we predicted $\mu_j^{(B)}$ using the estimated intercept term $\alpha_{s_j}^{(B)}$, the building-count-related covariate (Covariate A) $X_j^{(B)}$, and the estimated settlement-class specific effects $\beta_{s_j}^{(B)}$ for each settled grid cell $j$.

$$\mu_j^{(B)} = \alpha_{s_j}^{(B)} + \beta_{s_j}^{(B)} \cdot X_j^{(B)}$$

(11)

In Equation (12), we drew the predicted population densities $\lambda_j^{(D)}$ from a log-normal distribution, where $\mu_j^{(D)}$ is the predicted expected population density, and $\sigma_{s_j}^{(D)}$ represents the estimated settlement-class-specific variance term for each settled grid cell $j$.

$$\lambda_j^{(D)} \mid \mu_j^{(D)}, \sigma_{s_j}^{(D)} \; \sim \; \text{logNormal}\left(\mu_j^{(D)}, \sigma_{s_j}^{(D)}\right)$$
(12)

In Equation (13), we predicted $\mu_j^{(D)}$ using the estimated settlement-class-specific intercepts $\alpha_{s_j}^{(D)}$ and the four population-density-related covariates $X_j^{(D)}$ (Covariates B-E) for each settled grid cell $j$, and the corresponding estimated fixed effects $\beta^{(D)}$.

$$\mu_j^{(D)} = \alpha_{s_j}^{(D)} + \; X_j^{(D)} \; \cdot \; \beta^{(D)}$$
(13)

**Areal aggregations.** We aggregated the grid-cell level population count $P_j$ predictions across larger operational health boundaries, such as health zones and health areas [32], and the province boundaries approximated by the modelling environment. In Equation (14), $P_A$ represents the predicted population count for the area $A$, and the summation runs over all grid cells within it.

$$P_A = \sum_{j \in A} P_j$$
(14)

Since $P_A$ retains the posterior distributions of $P_j$, the aggregated estimates also allow for uncertainty quantification, conditioned on the observed data under the model assumptions. For this purpose, we assessed the posterior distributions using different summary statistics, such as the mean of the posterior distribution and lower and upper 95% CIs representing the 2.5th and 97.5th percentiles. We also computed a standardised measure of model uncertainty representing the difference between the upper and lower 95% CIs divided by the mean estimate.

**Model implementation.** We implemented the model using Stan (2.32.2) [33] with the R interface package rstan (2.32.6) [34] using three Markov chain Monte Carlo (MCMC) chains and 5,000 iterations per chain, with 500 warm-up iterations discarded. Standard diagnostics, such as the number of iterations that ended with divergence, the number of iterations that saturated the maximum tree depth, and the estimated Bayesian Fraction of Missing Information (EBFMI) for each MCMC chain, were complemented by an assessment of the Bulk Effective Sample Size across the model parameters using rank normalised draws and the visual inspection of trace plots. We also examined in- and out-of-sample goodness-of-fit using stratified leave-$p$-out cross-validation with an 80–20% split. For in- and out-of-sample predicted building and population counts, we extracted posterior means to evaluate bias (i.e., the mean of residuals), imprecision (i.e., the standard deviation of residuals), inaccuracy (i.e., the mean of absolute residuals), $R^2$ values (i.e., the squared Pearson correlation coefficient among the residuals) consistently with existing Bayesian hierarchical models for high-resolution population estimation [10]. We also computed mean absolute error (MAE—i.e., the average absolute difference between the predicted values and the actual values) and root mean squared error (RMSE—i.e., the square root of the average squared differences between predicted and actual values), together with the percentage of observations falling within the 95% CIs [35]. We also assessed spatial autocorrelation in the model residuals using global and local Moran's I tests [36].

## Ethics approval

This study was approved by the Faculty Ethics Committee of the University of Southampton (ERGO II 89997). Written consent was obtained by survey respondents.

## Results

### Model parameters

We developed a Bayesian model estimating population count at high spatial resolution in Kasaï-Oriental, DRC, integrating building and population counts retrieved from micro-census surveys. The model achieved a minimum Bulk Effective Sample Size of 2,270.96 across the parameters from both models, indicating reliable posterior estimates. In the first component, presented in Equations (2–5), we modelled building count leveraging the relationships between observed building count and those derived from settlement data (Covariate A) across different settlement classes. Fig 2 (top) shows that the model parameters vary substantially across settlement classes, although they maintain a similar direction and order of magnitude. Notably, variations in the intercept $\alpha_s^{(B)}$ suggest greater bias in the settlement data for rural settlements rather than urban ones. Rural settlements also exhibit a substantially higher variance $\sigma_s^{(B)}$ and a weaker covariate effect $\beta_s^{(B)}$,

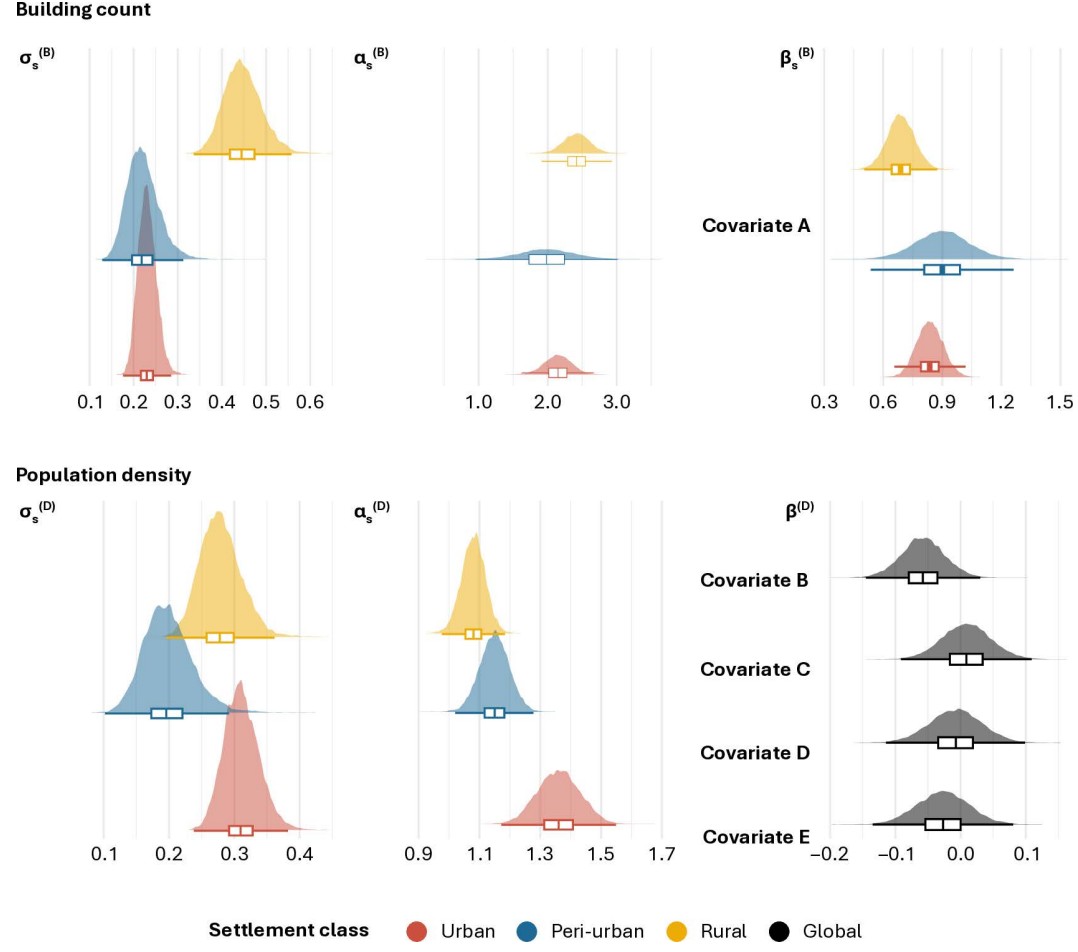

**Fig 2. Posterior density distributions and boxplots for the parameters estimated in the building count (top) and population density (bottom) model components on a logarithmic scale.** The intercept $\alpha$ and variance $\sigma$ are estimated independently by settlement class **s**. The multiplicative effects β are applied to the covariate building count from settlement data (A) by settlement class. Covariates B to E represent the fixed, global multiplicative effects for the covariates distance to main roads (B), average dry matter productivity (C), distance to violence against civilians (D), and terrain slope (E).

indicating looser relationships with observed building count. Peri-urban areas show a broader parameter spread, reflecting the heterogeneous nature of this transitional zone.

We modelled population density in a dedicated component presented in Equations (6–8). Fig 2 (bottom) shows that the population density parameters follow distinct yet consistent patterns across settlement classes. The intercept $\alpha_s^{(D)}$ suggests that mean population densities are higher in urban areas, followed by peri-urban, and rural areas, while variance $\sigma_s^{(D)}$ remains similar. Although covariate selection was based on correlations with log-population densities, the fixed multiplicative effects $\beta^{(D)}$ indicate minimal influence from most covariates. The strongest effect is observed for distance to main roads (Covariate B), where increased distance is associated with lower population densities. Average dry matter productivity (Covariate C) shows no distinct impact, likely due to varying relationships between population densities and ecosystems. Distance to violence against civilians (Covariate D) has a negative but weak influence on population density, potentially reflecting the push-pull effects of violence. Similarly, terrain slope (Covariate E) shows a negative but weak impact, suggesting that rugged relief is generally associated with lower population densities.

## Model fit

The model parameters presented above are used to estimate building and population counts within the 213 micro-census clusters. Fig 3 contrasts these estimates with the values observed in the respective clusters to provide a visual indication of the model goodness-of-fit. Both predicted building and population counts generally align with the respective observed values, with the 95% CIs typically overlapping the respective one-to-one relationship lines. Fig 3 (left) shows that observed building counts per cluster typically fall between 25 and 200 for all the settlement classes with a few outliers, particularly for clusters within rural settlements. In these clusters, the predicted building counts deviate from the one-to-one relationship line, but the 95% CIs successfully capture the observed building counts. As shown in Fig 3 (right), the range of observed population counts is wider, typically between 100 and 600 people per cluster. Outliers mainly emerge for higher population counts, reaching 1,050 people in one micro-census cluster. While the 95% CIs capture these extreme values, the mean predicted population counts consistently fall below the one-to-one relationship line. Distinct

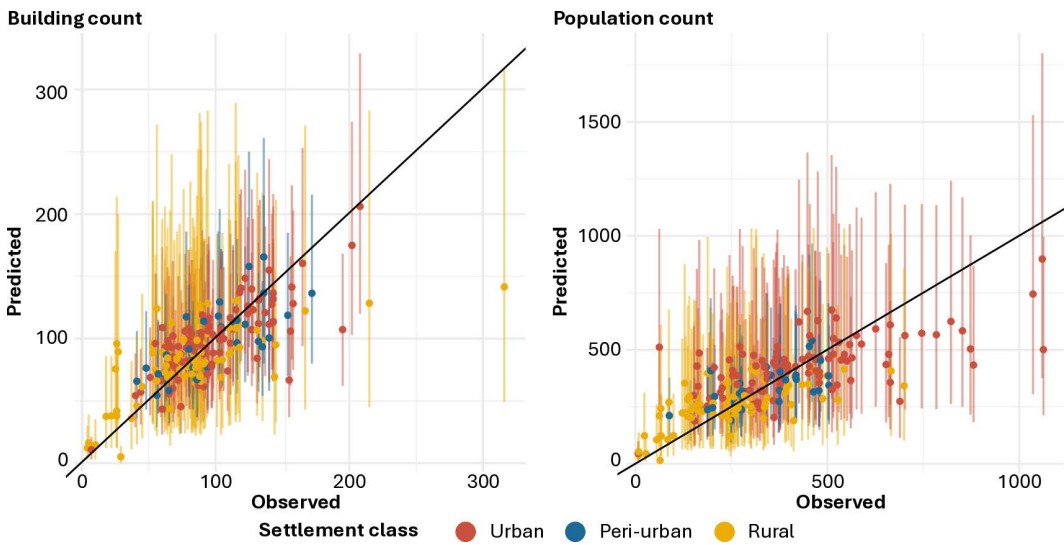

**Fig 3. Observed versus predicted building (left) and population (right) counts with 95% credible intervals (CIs) classified according to the respective settlement classes.** The diagonal black lines show a one-to-one relationship between observed and predicted values.

settlement-class-specific patterns also highlight the stronger impact of the hierarchical structure, compared to the covariates, as illustrated in Fig 2 (bottom).

Table 2 presents goodness-of-fit metrics for building and population counts based on the in-sample posterior predictions. The results show that, in both cases, approximately 95% of the observations fell within the respective 95% CIs indicating that the model structure is robust. Estimated building and population counts exhibit slight negative bias resulting in an average overprediction of 1.05 buildings and 2.24 people per cluster, respectively. Similarly, imprecision and inaccuracy are larger for estimated population counts than for estimated building counts. Both estimates show an analogous $R^2$ value of 0.52, indicating that slightly over 50% of the variance in the observed building and population counts is explained by the respective mean estimates. The residuals for both building and population counts exhibited significant ($p < 0.05$) but reduced spatial autocorrelation ($I = 0.15$) using the Moran's I test carried out for different distance classes. This result was observed only for the smallest distance class (0.05) and linked to systematic model underpredictions in three remote clusters located in the western part of the province. This analysis produced similar results for cross-validated out-of-sample predictions, with detailed results available on a public Zenodo repository [37], suggesting the absence of model overfitting. These results are presented in two separate files for in-sample (*_full) and out-of-sample (*_crossValidation) data in the directory population_model/model_evaluate.

## Grid-cell predictions and areal aggregations

Grid-cell-level model predictions were performed by linking the model parameters presented above with the covariates and settlement classes derived from the geospatial gridded datasets. This procedure resulted in 13,500 posterior predictions for each of the 81,446 settled grid cells in the modelling environment. Fig 4 shows the geographic distribution of the mean gridded population estimates across Kasaï-Oriental (left) with a zoom-in of the provincial capital of Mbuji-Mayi (right). The figure shows well-defined geographic patterns informed by the distribution of the settled grid cells. In most of the province, the estimated population counts per grid cell are low, with values generally below 100 people per grid cell. However, these values tend to be higher in urban areas, particularly in Mbuji-Mayi, with mean estimated values reaching up to 488 people per grid cell. The population patterns across health zones and health areas show notable differences. For instance, the health zone of Kabeya Kamwanga, located in the predominantly rural eastern part of the province, covers approximately 2,579 km², with a large but scattered estimated

**Table 2. Analysis of residuals for building and population counts based on in-sample posterior predictions.**

|  | Building count | Population count |
|---|---|---|
| **Bias** | −1.05 (−0.12) | −2.24 (−0.28) |
| **Imprecision** | 27.52 (0.44) | 133.38 (0.95) |
| **Inaccuracy** | 19.47 (0.27) | 98.76 (0.47) |
| ***R²*** | 0.52 | 0.52 |
| **RMSE** | 27.48 | 133.09 |
| **MAE** | 19.47 | 98.76 |
| **Observed within 95% CIs** | 96.24% | 93.43% |

Bias represents the mean of the residuals, imprecision the standard deviation of residuals, inaccuracy the mean of absolute residuals, $R^2$ the squared Pearson correlation coefficient among the residuals, and the percentage of observations falling within the 95% credible intervals (CIs). Residuals are computed as the observed values minus the posterior means. The metrics in parentheses are standardised using scaled residuals (residuals/posterior means).

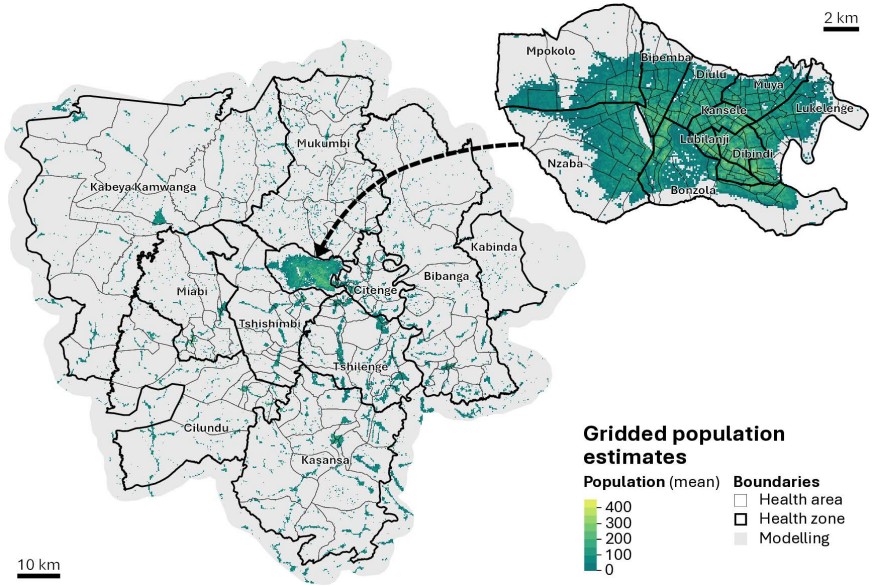

**Fig 4. Mean gridded population estimates across Kasaï-Oriental (left) with a zoom-in of the provincial capital of Mbuji-Mayi (right).** Fully overlapping health zones (labelled) and health areas (unlabelled) are overlayed. Administrative boundaries sourced from OpenStreetMap (https://planet.osm.org) under the Open Database License (ODbL) (https://www.openstreetmap.org/copyright). Health zone and health area boundaries sourced from GRID3 (https://data.grid3.org/) under the CC BY 4.0 license (https://creativecommons.org/licenses/by/4.0/).

population. Conversely, the health zone of Lubilanji, in Mbuji-Mayi, covers an area of 4 km², with a smaller but densely distributed estimated population.

Fig 5 provides probabilistic insights into the population of the province, as well as its health zones and health areas resulting from the aggregation of the grid cell posterior distributions. The size of the largest circle indicates the mean population estimate for Kasaï-Oriental, as approximated by the modelling environment, which is likely to count 4.1 million people, with the respective 95% CIs with a 95% probability that the population is between 3.4 and 4.8 million, resulting in an uncertainty level of 34.30%. Health-zone aggregates confirm that the most populated, namely the Kasana (434,113 mean population estimate) and Kabeya Kamwanga (411,389 mean population estimate) health zones, feature the largest geographic extent. The opposite is generally true for the smallest health zones in Mbuji-Mayi, with the notable exception of the Kabinda health zone (22,960 mean population estimate) in the eastern part of the province, which is partly falling outside of the modelling environment. Fig 5 also shows how uncertainty varies across levels of aggregation, from the health area to the health zone and the province, with uncertainty levels typically increasing with granularity. For instance, the Tshilenge health zone exhibits an overall uncertainty level of 46.7%, with its health areas exceeding 46.67% and reaching a maximum of 148.52%. Other health zones feature more complex uncertainty patterns underscoring the need for individual assessments of the posterior distribution aggregates.

The input data and scripts used to develop the modelled gridded population estimates, as well as the grid-cell posterior distributions and scripts to aggregate them within user-defined geographic boundaries are available on a public Zenodo repository [37].

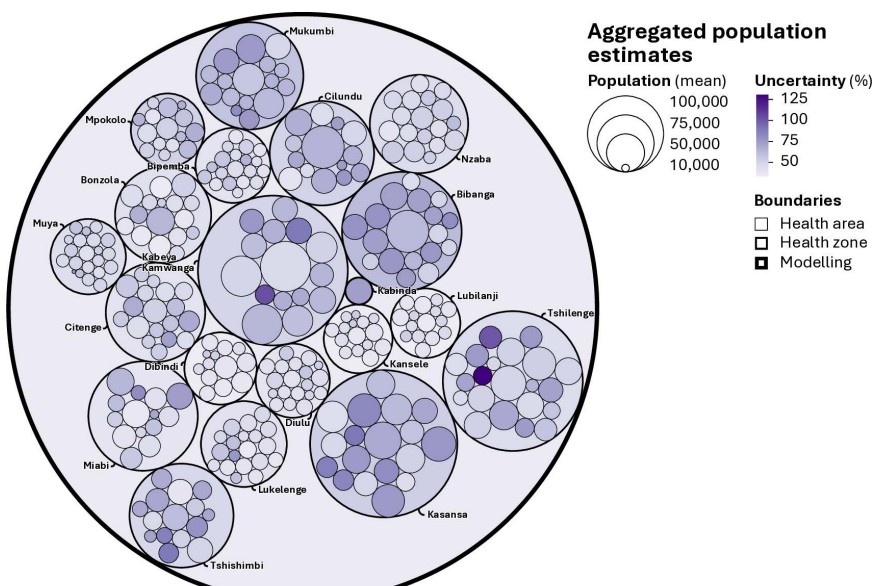

**Fig 5. Mean population count estimates and uncertainty levels for Kasaï-Oriental, its health zones (labelled) and health areas (unlabelled).** The size of the circles represents the mean population estimate while the colour scale represents the uncertainty level computed as the difference between the upper and lower 95% CIs divided by the mean estimate.

## Discussion

This study introduced a Bayesian model jointly estimating building and population counts at high spatial resolution in the Kasaï-Oriental province, DRC. The model integrated building and population counts from micro-census data with geospatial datasets to estimate population counts within approximately one-hectare grid cells across the province. It enhanced existing Bayesian population models [10,11,13] by explicitly addressing uncertainties in building footprint data, a previously unaccounted-for source of error [11]. Consistent with earlier modelling efforts in the DRC and other countries in sub-Saharan Africa, the results underscore the advantages of a hierarchical model structure, where key parameters— such as variance, intercept, and covariate effects—are estimated for each settlement class [10,11,13]. In prior studies, this approach enabled to successfully estimate population counts at high spatial resolution beyond the surveyed regions [10].

The model demonstrated strong goodness-of-fit, showing minimal bias, imprecision, and inaccuracy compared to similar Bayesian population models [10,11,13]. However, we detected significant but limited spatial autocorrelation in the model residuals using Moran's tests, which is unlikely to lead to substantial underestimation of uncertainty in aggregate estimates. A key innovation was the building count model component, driven by covariate building count from settlement data and parameters estimated by settlement class, underscoring greater bias in settlement data for rural areas compared to urban areas [38]. This previously unaddressed source of uncertainty was successfully integrated into the population count model component implemented in other modelling work in the DRC [11,21]. In this component, the impact of the parameters is mostly associated with the settlement-class parameters rather than the effects of the covariates related to population density, similar to existing studies [10]. Because these covariates explain relatively small additional variation beyond the settlement-class specific intercepts, the posterior estimates of population density tend to shrink toward those intercepts. In this regard, our analysis underscored the limited number of gridded datasets exhibiting significant correlations with log-population densities, emphasising the need for further research into alternative spatial datasets in data-scarce contexts. As a result of this shrinkage towards settlement-class specific population densities, predicted population counts feature reduced variability, particularly in urban areas. However, approximately 95% of the observed building and population counts fell within the respective 95% CIs.

Central to this modelling effort was the availability of actionable micro-census survey data. The definition and selection of these small areas were designed to provide an accurate and unbiased representation of Kasaï-Oriental's population, while capturing differences in demographic characteristics and context. The data were collected in approximately two months, arguably, at a fraction of the cost of a complete enumeration. Moreover, the insecurity and natural hazards that affected the logistical implementation of the micro-census survey underscore the challenges of carrying out complete enumeration over larger areas, particularly those that are difficult to access. Although a similar micro-census survey is unlikely to be replicated in the near future, the integration of routinely collected health data [39] or available digital traces [40] could enable subsequent model updates capturing population dynamics also across unsurveyed provinces [10].

The flexible aggregation of the gridded population estimates with operational public health boundaries demonstrated the ability to transfer uncertainty horizontally across health areas and vertically between health areas and health zones. These aggregations are made possible by the consistency of the gridded framework, which effectively addresses spatial misalignment between administrative and operational health boundaries [41]. At the province level, the 34.30% uncertainty around the 4.1 million mean population estimate encompassed the available population sources for the Kasaï-Oriental province, including the governmental estimates of 3.4 million for 2021 [14]. The results confirmed that uncertainty typically increases with granularity [10], with an exceptional uncertainty level of 148.52% detected in a health zone with a mean population estimate of 10,949, and 95% CIs ranging from 5,026–21,289. These health zone and health area estimates provide an opportunity to inform probabilistic scenarios, enabling the public health community to improve planning and intervention [15,16]. For example, upcoming polio vaccination campaigns in Kasaï-Oriental could benefit from allocating resources based on the upper 95% credible intervals to ensure optimal coverage.

This study highlights the potential of Bayesian models for estimating population counts at high spatial resolution in data-scarce settings where uncertainties in available demographic datasets can be substantial [2]. However, several critical challenges must be addressed to foster the adoption of these methods within the public health community. First, future work should focus on building the necessary skills and technical capacity to understand, implement, and interpret Bayesian models effectively [3]. While the required effort varies by context, recent initiatives have demonstrated notable successes, with the Colombian National Administrative Department of Statistics successfully generating Bayesian population estimates for hard-to-reach regions of the country [42]. Second, Bayesian posterior distributions should be integrated with relevant ancillary datasets, such as existing population pyramids, to estimate the sizes of critical population subgroups (e.g., under-one and under-five) [11]. Lastly, operational relevance should be improved by expanding existing tools allowing for the seamless application of probabilistic approaches to gridded population estimates and area aggregates in health system planning [43]. Ensuring that Bayesian high-resolution population estimates are accessible, relevant, and easy to use will be critical to addressing public health data gaps and supporting accurate planning and intervention.

## Supporting information

**S1 Table. Gridded Geospatial Datasets.** The table provides an overview of the gridded geospatial datasets tested in the population count model. The Name column provides the dataset name, Description a succinct description, Year indicates the reference year, Summary outlines the type of processing applied, and Source identifies the dataset origin. All datasets have been processed by WorldPop at the University of Southampton.
(PDF)

## Acknowledgments

We acknowledge the support of partner organisations, including the Center for Integrated Earth System Information (CIESIN) at the Columbia Climate School at Columbia University, the École de Santé Publique de Kinshasa, the Flowminder Foundation, GRID3 Inc., and WorldPop at the University of Southampton, in the completion of this study. We also

acknowledge the critical contribution of the teams of surveyors associated with the École de Santé Publique de Kinshasa involved in the fieldwork.

## Author contributions

**Conceptualization:** Gianluca Boo.

**Data curation:** Gianluca Boo, Edith Darin, Heather R. Chamberlain, Roland Hosner, Pierre K. Akilimali, Henri Marie Kazadi.

**Formal analysis:** Gianluca Boo.

**Funding acquisition:** Andrew J. Tatem.

**Investigation:** Gianluca Boo.

**Methodology:** Gianluca Boo, Edith Darin.

**Project administration:** Heather R. Chamberlain, Roland Hosner, Pierre K. Akilimali, Attila N. Lázár.

**Validation:** Edith Darin, Heather R. Chamberlain, Roland Hosner, Chibuzor C. Nnanatu.

**Visualization:** Gianluca Boo.

**Writing – original draft:** Gianluca Boo.

**Writing – review & editing:** Gianluca Boo, Edith Darin, Heather R. Chamberlain, Roland Hosner, Pierre K. Akilimali, Henri Marie Kazadi, Chibuzor C. Nnanatu, Attila N. Lázár, Andrew J. Tatem.

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
