## [Decision Letter · Decision Letter 0]

24 Feb 2025

PGPH-D-24-02971

Tackling public health data gaps through Bayesian high-resolution population estimation: a case study of Kasaï-Oriental, Democratic Republic of the Congo

Dear Dr. Boo,

Thank you for submitting your manuscript to PLOS Global Public Health. After careful consideration, we feel that it has merit but does not fully meet PLOS Global Public Health’s publication criteria as it currently stands. Therefore, we invite you to submit a revised version of the manuscript that addresses the points raised during the review process.

Please see reviewers' comments below, including the attached file. 

We look forward to receiving your revised manuscript.

Kind regards,

Mark C. Wheldon, Ph.D.

Academic Editor

Journal Requirements:

Additional Editor Comments (if provided):

Reviewer 3 noted that the link at the start of the manuscript pdf file leads to a dead site. Please make sure that the source data are publicly available, or explain why this isn't possible.

Reviewers' comments:

Reviewer's Responses to Questions

**Comments to the Author**

1. Does this manuscript meet PLOS Global Public Health’s publication criteria?

Reviewer #1: Yes

Reviewer #2: Yes

Reviewer #3: Yes

2. Has the statistical analysis been performed appropriately and rigorously?

Reviewer #1: I don't know

Reviewer #2: I don't know

Reviewer #3: Yes

3. Have the authors made all data underlying the findings in their manuscript fully available (please refer to the Data Availability Statement at the start of the manuscript PDF file)?

Reviewer #1: Yes

Reviewer #2: Yes

Reviewer #3: No

4. Is the manuscript presented in an intelligible fashion and written in standard English?

Reviewer #1: Yes

Reviewer #2: Yes

Reviewer #3: Yes

Reviewer #1: I found the topic of the paper to be interesting and the goal of joint modelling with rigorous assessment of uncertainty to be good. However, I have some issues with the presentation and I feel that it needs to become more clear.

1. Abstract: "The model exhibits limited bias, inaccuracy and imprecision..." This is a imprecise statement since it is subject to subjective assessment of what "limited" means. The authors should be more precise so that the readers of the abstract has an idea about the amount of bias and uncertainty.

2. Lines 87--89: This mitigation was very unclear to me. What does it mean to extend the "modeling environment". Are the authors calculating the population of a larger geographical area than intended? Clarification is needed.

3. Lines 92--109: Not all readers will be familiar with the micro-census data. I would appreciate more information. How were the micro-census clusters selected? Are they of uniform size?

4. Lines 111--138: There is too little information about the 292 datasets. Why exactly 292 datasets? Were transformations applied to the 292 datasets? Why were they reduced to exactly 5 datasets? What was the criteria for inclusion/non-inclusion? How large are the micro-clusters compared to the approximately 100 m x 100 m grid cells?

5. Line 135: Minor comment. I would replace "the mean" with "the average" and "standard deviation" with "empirical standard deviation" (since these are just summary quantities for the rasters)

6. Lines 140--: The statistical model is confusing to me. A data model is described for observed clusters, but how does it extend to unobserved grid cells? It would be very beneficial to first formulate the relationship between mean and covariates at the grid cell level (so that it is defined for all grid cells), and then describe how the micro-census data informs about the grid cell means.

7. Lines 140--: The use of the word Poisson process needs more description. It was very confusing to me. The index i indexes micro clusters so are the authors claiming that this is a Poisson process on micro clusters? Maybe what the authors intended is that B_i (where i indexes cells) is a discretized Poisson process where the intensity is given by log(\mu_i) = covariates+random effect? Then each microclusters j = 1,..., 213 corresponds to cells i[j] (???) which is then observed exactly.

8. Lines 140--: There is a need to write something like s[i] to indicate that settlement class is determined by the grid cell.

9. Lines 151: It is claimed that \hat{B}_i is on logarithmic scale, but this cannot be true?

10. Line 157: It might be better to write LogNormal than log\mathcal{N} since the latter could be confused with the logarithm of a normal distributed variable (which does not make sense).

11. Lines 164--178: The same comments are valid here as in points 6.--10.

12. Line 168: The "." should be a "\cdot".

13: Line 177: I don't understand the discussion about "random effects". Isn't the description in Equation (5) implicitly the inclusion of a random effect in the log-linear model?

14. Lines 140--: NB. The authors present a Bayesian model. Then there is a need to be careful with the conditioning in the equations. E.g., Equation (1) is wrong. It is B_i | \hat{B}_i which is Poisson(\hat{B}_i). The same is true in Equation (2), you need to condition on \mu_i^(B) and \sigma_s^(B) for this to be logNormal. Please check all the equations.

15. Line 192: "5000 iterations" and "500 warm-up iterations" are not useful in it itself without knowing anything about the mixing properties of the Markov chain. Could the authors provide some metric like effective sample size to describe how informative 5000 iterations are?

16. Lines 207--208: This sentence was unclear to me since the model is explicitly defined on the micro-census clusters. How can it be extended? If you describe the model more clearly, I believe this problem will go away. But I would be interested in a clarification about whether the "random effect"/logNormal is used also for unobserved grid cells?

17. Line 212: Minor comment. I would write "... conditioned on the observed data under the model assumptions". It is a bit imprecise to state that you condition on the model assumptions.

18. Line 217: It would be nice for the reader to elaborate on what is meant by "comparable" measure.

19. Lines 295--297: This was confusing to me. You only fit the joint model, right? Then what does it mean that "the population count model does not noticeably improve the ability to capture variability in the observed data"?

20. Table 2: Why is in-sample properties interesting? The model includes a random effect so in-sample properties might be very different from out-of-sample properties?

21. Lines 313--314: How were different age and sex classes suddenly introduced? There is nothing in the described model about this. The authors need more details here.

22. Line 359: I find the statement "novel Bayesian model" to vague. Novel in which sense?

23. Discussion: A concern for me would be that there is an assumption that the "residuals" from the mean in each grid cell are iid. I would believe there is considerable spatial structure in this residuals. The consequence of ignoring this would typically be to underestimate uncertainty in the aggregate estimates (for larger areas). I think this should be discussed. There is no out-of-sample evalutions to assess this potential issue either.

Reviewer #2: 1. The paper addresses an important topic and a real challenge in low resource settings. Thank you for working on this.

2. I am not a statistician, and I cannot comment on the quality and novelty of the model / approach, nor on the extent to which it is correct. I would therefore defer to a statistician for this appraisal.

3. I accepted to review the paper because I was expecting more implications for public health, given that the journal is Plos Global Health.

I would suggest clarifying and discussing points such as:

i) The complexity to estimate population for health areas/ zones given that they differ from the DRC administrative divisions.

ii) The feasibility for national authorities to learn from / replicate / update this approach for policy making, and the related resources and skillset needed.

iii) The feasibility, cost, and duration related to the implementation of the micro-census.

iv) The need for and the challenges related to updating the population estimates and include population displacement (should the region be affected again by insecurity, or should the model be replicated in other provinces of the DRC).

4. Furthermore, I would recommend clarifying why you chose to present the results on Kasai-Orientale, given that, if I understand correctly, several estimates have been produced with/ through the collaboration with GRID3 (https://grid3.org/geospatial-data-drc). It would be helpful to clarify whether the Kasai-Oriental approach is different in some ways and what the link with GRID3 is. You do mention GRID3 in the financial disclosure, but I would suggest explaining it in the paper itself.

5. While the lack of population estimates is big problem, it is not the only challenge to health service delivery in low-resource settings. I would recommend nuancing the introduction to recognize that lack of resources, limited access to population or health facilities, insecurity, logistical constraints, etc are major challenges that would persist even with reliable population estimates.

Reviewer #3: The manuscript focuses on creating and using a Bayesian hierarchical model to estimate population counts by levering a building count model and a population count model. This work is part of the greater goal to provide reliable population size estimates to organizations who require size estimates to implement health improvement measures.

I enjoyed reading this manuscript and recognize the importance of population size estimation in the context of improving public health. While I have no concerns about the scope and motivation behind this work, I have several comments regarding the statistical modeling sections, which I enumerate below. Please see attached review.

**Do you want your identity to be public for this peer review?** For information about this choice, including consent withdrawal, please see our Privacy Policy

Reviewer #1: No

Reviewer #2: No

Reviewer #3: No

---

## [Decision Letter · Decision Letter 1]

26 Jun 2025

PGPH-D-24-02971R1

Tackling public health data gaps through Bayesian high-resolution population estimation: a case study of Kasaï-Oriental, Democratic Republic of the Congo

Dear Dr. Boo,

Thank you for submitting your manuscript to PLOS Global Public Health. After careful consideration, we feel that it has merit but does not fully meet PLOS Global Public Health’s publication criteria as it currently stands. Therefore, we invite you to submit a revised version of the manuscript that addresses the points raised during the review process.

Please see additional comments from Reviewer 3.

We look forward to receiving your revised manuscript.

Kind regards,

Mark C. Wheldon, Ph.D.

Academic Editor

Additional Editor Comments (if provided):

Reviewers' comments:

Reviewer's Responses to Questions

**Comments to the Author**

Reviewer #1: All comments have been addressed

Reviewer #3: (No Response)

publication criteria?

Reviewer #1: Yes

Reviewer #3: Partly

3. Has the statistical analysis been performed appropriately and rigorously?

Reviewer #1: Yes

Reviewer #3: I don't know

4. Have the authors made all data underlying the findings in their manuscript fully available (please refer to the Data Availability Statement at the start of the manuscript PDF file)?

Reviewer #1: Yes

Reviewer #3: Yes

5. Is the manuscript presented in an intelligible fashion and written in standard English?

Reviewer #1: Yes

Reviewer #3: No

Reviewer #1: (No Response)

Reviewer #3: Please see attached document.

**Do you want your identity to be public for this peer review?** For information about this choice, including consent withdrawal, please see our Privacy Policy

Reviewer #1: No

Reviewer #3: No

---

## [Decision Letter · Decision Letter 2]

1 Aug 2025

Tackling public health data gaps through Bayesian high-resolution population estimation: a case study of Kasaï-Oriental, Democratic Republic of the Congo

PGPH-D-24-02971R2

Dear Mr. Boo,

We are pleased to inform you that your manuscript 'Tackling public health data gaps through Bayesian high-resolution population estimation: a case study of Kasaï-Oriental, Democratic Republic of the Congo' has been provisionally accepted for publication in PLOS Global Public Health.

Best regards,

Mark C. Wheldon, Ph.D.

Academic Editor

Reviewer Comments (if any, and for reference):

Reviewer's Responses to Questions

**Comments to the Author**

Reviewer #3: All comments have been addressed

publication criteria?

Reviewer #3: Yes

3. Has the statistical analysis been performed appropriately and rigorously?

Reviewer #3: Yes

4. Have the authors made all data underlying the findings in their manuscript fully available (please refer to the Data Availability Statement at the start of the manuscript PDF file)?

Reviewer #3: Yes

5. Is the manuscript presented in an intelligible fashion and written in standard English?

Reviewer #3: Yes

Reviewer #3: I thank the authors for thoughtfully addressing all of my comments.

**Do you want your identity to be public for this peer review?** For information about this choice, including consent withdrawal, please see our Privacy Policy

Reviewer #3: No
